# Optimal Relay Selection Scheme with Multiantenna Power Beacon for Wireless-Powered Cooperation Communication Networks

**DOI:** 10.3390/s21010147

**Published:** 2020-12-29

**Authors:** Oussama Messadi, Aduwati Sali, Vahid Khodamoradi, Asem A. Salah, Gaofeng Pan, Shaiful J. Hashim, Nor K. Noordin

**Affiliations:** 1Wireless and Photonic Networks Research Centre of Excellence (WiPNET), Department of Computer and Communication Systems Engineering, Faculty of Engineering, Universiti Putra Malaysia, Serdang 43400, Selangor, Malaysia; v.khodamoradi.86@gmail.com (V.K.); sjh@upm.edu.my (S.J.H.); nknordin@upm.edu.my (N.K.N.); 2Department of Computer System Engineering, Faculty of Engineering and Information Technology, Arab American University, Jenin, West Bank, Palestine; Asem.Salah@aauj.edu; 3Computer, Electrical and Mathematical Sciences and Engineering Division, King Abdullah University of Science and Technology (KAUST), Thuwal 23955-6900, Saudi Arabia; gaofeng.pan@kaust.edu.sa

**Keywords:** energy harvesting, power-beacon, cooperative-networks, relay selection, multi-antennas, beamforming

## Abstract

Unlike the fixed power grid cooperative networks, which are mainly based on the reception reliability parameter while choosing the best relay, the wireless-powered cooperative communication network (WPCCN) and in addition to the reception reliability the transmission requirement consideration is important for relay selection schemes. Hence, enabling efficient transmission techniques that address high attenuation of radio frequency (RF) signals according to the distance without increasing the total transmission power is an open issue worth studying. In this relation, a multiantennas power beacon (PB) that assists wireless-powered cooperative communication network (PB-WPCCN) is studied in this paper. The communication between source and destination is achieved with the aid of multiple relays, where both the source and the multiple relays need to harvest energy from the PB in the first place to enable their transmission functionalities. A novel relay selection scheme is proposed, named as two-round relay selection (2-RRS), where a group of relays that successfully decode the source information is selected in the first round selection. In the second round, the optimal relay is selected to forward the recorded information to the destination. The proposed 2-RRS scheme is compared with two existing relay selection schemes, i.e., partial relay selection (PRS) and opportunistic relay selection (ORS). The analytical closed-form expressions of outage probability and average system throughput are derived and validated by numerical simulation. The comparison results between different relay selection schemes show: (I) The superiority of the proposed 2-RRS scheme as it achieves around 17% better throughput compared to the conventional ORS scheme and 40% better than the PRS scheme, particularly when PB transmit power is 10 dB; (II) The proposed 2-RRS scheme guarantees the lowest outage probability, especially when the PB is equipped with multiantennas and performs beamforming technique; (III) The optimal localisation of the PB between the source and *N* relays depends on the adopted relay selection scheme; (IV) The exhaustive search of the maximum system throughput value shows that the proposed 2-RRS scheme required shorter energy harvesting time compared to other schemes. The increase in energy harvesting time and number of relays do not necessarily reflect positively on the system throughput performance; hence tradeoffs should be taken into consideration.

## 1. Introduction

### 1.1. Background

The energy harvesting (EH) is a promising technique for enabling green communication systems. It helps terminals recharge their batteries from the surrounding environment, including wind, solar, vibration, and thermoelectric, to convert them into electrical energy [1]. However, these natural renewable sources of energy are characterized by the lack of permanence and reliability, such as clouds obscured sunlight or the absence of wind. Therefore, wireless communication systems using conventional energy harvesting sources cannot ensure sustainability and QoS requirements. Besides, in some scenarios, wireless nodes may require frequent charging or a replacement battery. They may not be accessible such as in devices hidden (e.g., in the walls, bodies or appliances) or can be deployed in toxic areas and hazardous industrial environments (e.g., in pressurized pipes or radioactive areas) that require high operating costs.

Radio-frequency (RF) signals were mainly used as a means of transmitting information in wireless communications. Varchney suggested that the RF signals can be used to transmit information and provide energy at the same time [2]. This potentiality has opened a new wireless-powered communication (WPC) research paradigm, in which wireless terminals harvest RF signals radiated by dedicated power transmitter(s), and the harvested energy is used for transmission or processing [3]. WPC has become an alternative technology to power the next generation of green wireless communications. However, the high attenuation of RF signals according to the distance between the transmitter and receiver, in addition to the maximum transmit power limitation due to the safety issues makes WPC applications more effective in wireless network topology that requires low-power to enable its functionality, such as RFID tags [4] and sensor networks [5].

Present research interested in enabling the WPC in massive MIMO system focuses primarily on beamforming schemes design, where a portion of time slot was reserved for energy transfer and information transmission [6,7,8,9]. Moreover, in cooperative communication and relay selection, the distances between transmitter and receiver will significantly reduce, which made wireless power cooperative communication networks (WPCCN) an important technology that can efficiently combat signal propagation losses for information transmission and wireless energy [10,11,12]. Therefore, by virtue of the aforementioned emerging technologies, RF energy harvesting efficiency can be significantly increased, and signal degradation will be less severe. It can be expected that WPCCN will become an essential and important technology in the near future for many industrial and commercial systems, including the wireless sensor networks and the emerging Internet of Things (IoT) [13,14,15,16].

### 1.2. Related Works

In most of the literature on the topic of WPCCN, a three-node configuration is considered. Due to long-distance and high path loss between, an EH relay node cooperates to enable communication between source and destination nodes. We can group the WPCCN related works into two main clusters based on the number of relay(s), namely: (1) Single-relay WPCCN (2) Multirelays WPCCN.

#### 1.2.1. Single-Relay WPCCN

The authors in [17] analyzed the harvest-then transmission (HTT) protocol. Firstly, the receiver users harvest energy through the downlink from the access point (AP). Then, in the uplink they send their information to the AP using multiple access time-division (TDMA). The HTT protocol was extended in [11] to harvest-then-cooperate (HTC) protocol, where a source and a single relay have to harvest energy from an access point (AP) on the first time slot, then act collaboratively to transfer information from the source to the destination. In [18] to successfully deliver the information toward the destination the simultaneous wireless information and power transfer (SWIPT) technique is introduced, where the source transmits RF signal the single relay node must switch between data transmission and energy harvesting. In [19] a simple cooperative system setup is considered, where a single source, relay, and destination communicate via Rayleigh fading channel, based on time switching (TS) and power splitting (PS) protocols. The single relay harvests the RF signal and applies amplify-and-forward (AF) relaying to deliver source information to the destination: first, the system throughput in this work was derived using two transmission modes. First, a delay-limited transmission mode, where the system outage probability is evaluated to determine the system throughput with fixed source transmission rate. Second, the delay-tolerant transmission mode where the source can transmit until the evaluated ergodic capacity. The latter is what determines the system throughput in this mode. Multisource single relay and destination in decode-and-forward (DF) protocol was studied in [12], where the destination and multisource nodes communicate in full-duplex Rayleigh fading channels with and without the help of the single relay. The time interval was divided into two equal slots. The relay node working with the power splitter technique (PS) where the received signal from the multiple sources in the first time slot T/2 is divided into two portions. The first portion of sources’ signal received at the relay is allocated to energy harvesting. The second portion is reserved for information decoding and forwarding from the relay to destination in the remaining half-interval time slot T/2. The same authors in [20] used WPCCN system model configuration as in [12]. However, in [20] considered multidestination and single relay instead of multisource and single relay. A user selection is introduced over Nakagami-m and Rayleigh channels.

The deficiency lies in WPCCN with single relay configuration, when the energy harvesting from the source is not enough for the single relay to charge its battery and forward the decoded information to the destination due to path loss and shortage in energy. The idea of placing a dedicated wireless power transmitter known as power beacon (PB) to support wireless communication systems and help terminals wirelessly do charging via the RF power transfer was suggested [21]. The deployment of dedicated PB in an existing WPCCN such in [22,23] has been designed in a way that the network modified can enhance the wireless charging of terminals and wireless access. The authors in [22] propose a power beacon-assisted both the source and the single relay to harvest the RF energy from the PB and enable communication between the source and the destination. The outage probability of the system was derived over a Rayleigh fading channel. In [23] the PB assist both the source and the single relay also with enabling beamforming technique using the multiantennas at the PB. The outage performance was analyzed over a multiuser nonorthogonal multiple access (NOMA) Nakagami-m fading channels. The results have shown that increasing PB antennas number and enabling sharp energy beamforming will significantly improve the system’s outage performance.

#### 1.2.2. Multirelays WPCCN

In practice, there may exist more than one relay that has the possibility to assist the communication between the source and the destination. Most related works on the multirelay WPCCN have been considered relay selection schemes for cooperative communications. To achieve full cooperative diversity gains, relay selection techniques were suggested as an effective solution to overcome fading and system outage while sustaining spectral efficiency. Opportunistic relay selection (ORS) and partial relay selection (PRS) are the two most common relays among various relay selection schemes [11,24,25,26,27,28].

ORS and PRS schemes’ mean difference is that the channel state information (CSI) of the two hops (S-R link and R-D link) is considered to select the best cooperative relay in the ORS scheme. However, in PRS, it is assumed that only the CSI of one hop is available during selection [26]. In [24] the PRS scheme was used to select a single relay based on CSI of the S-R Link only. Comparing to the ORS scheme, the PRS scheme can prolong the lifetime of the energy-constrained relay node. However, the performance of PRS is limited due to partial channel knowledge which cannot fully represent the end-to-end channel quality. The authors in section IV [11] extend the single-relay analysis to a multirelays scenario where ORS and PRS schemes are considered and compared. The source and relays set have to harvest energy from AP on the first time slot. The best relay maximized the end-to-end SNR is selected with ORS scheme to act collaboratively and transfer information from the source to the destination. In PRS the best relay selected based on CSI of one hope only. The numerical results show the superiority of the ORS scheme over the PRS scheme. A Dual-hop DF cooperative network is considered in [25] where the multirelays enable energy harvesting in the first stage to support the information decoding and forwarding at the selected relay. The ORS and PRS relay selection schemes are utilized to increase the diversity gain and decrease the outage probability. Over independent Rayleigh fading channels, the ORS and PRS outage probabilities were analyzed. The authors have shown with numerical results that ORS scheme overtakes PRS scheme in terms of performance. However, on the other hand, ORS also increases the cooperative overhead compared to PRS.

In [26] a PB-assisted cognitive radio wireless sensor network is considered. Both source and relay nodes must harvest energy from the multiantenna PB to enable their information transmission using the DF protocol. The authors proposed the ORS and PRS relay selection scheme to overcome spectrum scarcity and energy constraints for cognitive radio. Exact and asymptotic formulas of the system throughput and outage probability are derived with Rayleigh fading channel. A PB-assisted WPCCN using NOMA is considered in [27]. A new PRS scheme is introduced for two cases: when the far user can only receive the signal from relays, and when both the distant and the near users can receive signals from source and relays. The analytical expressions for the near and far user were derived over Nakagami-m channel. In [28] the outage performance of WPCCN system model has been investigated. A threshold for DF protocol is considered for and ORS and PRS schemes. The authors defined two PRS subschemes as follows: when the CSI of S-R link is available, they called this subscheme OSRL-WEH and when the CSI of S-R link is not available, and that of R-D link is available, namely ORDL-WEH. Their numerical results have shown the positive impact of increasing the number of relay candidates on the system performance. ORS outperform both PRS schemes due to full CSI knowledge. ORDL scheme outperforms OSRL scheme due to his final overall OP in better partial CSI. A four-node system model topology has been recognized previously by our research group [29], in which the power beacon assisted a wireless power cooperative communications network (PB-WPCCN). Both the relay terminals and the source have no connection to the power grid, and they need to harvest energy from the RF signals radiated by the PB and work together to forward the information to the destination successfully.

Considering a wireless- powered cooperative communication network (WPCCN) harvests energy from a dedicated power-beacon source is highly poses the development of new cooperative relay selection schemes. ORS and PRS schemes have mainly been developed to suit fixed power grid cooperative networks, which are mainly based on the reception reliability parameter only while choosing the best relay, the WPCCN, and the reception reliability the transmission requirement consideration is important for relay selection schemes [30]. Thus, the max-min criterion will not be the diversity-optimal in WPCCN, since the best relay for energy harvesting does not necessarily coincide with the preferable relay for information transmission. To improve the performance of the studied PB-assisted WPCCN system, the relay selection criteria should take into consideration the best channel condition for both the reception reliability and the transmission power, which is fundamentally different from conventional relay selection schemes aforementioned. In this paper, a new relay selection scheme was introduced and compared with conventional schemes. A practical WPCCN protocol is considered known as time switching harvest then cooperate protocol (TS-HTC). In this protocol, the source and the relays spend some time for energy harvesting from the PB and use the remaining time slot for information processing.

The contributions of this work are summarized in the following:A multiantennas PB-WPCCN system model is proposed, where both the source and the relays set are assumed not connected to a fix power grid. Alternatively, they need to harvest energy first from the dedicated multiantennas PB and then work cooperatively to forward the information to the destination.A two-stage relay selection scheme is introduced, referred to as two-round relay selection (2-RRS) and compared with two popular relay selection schemes, i.e., partial relay selection (PRS) and opportunistic relay selection (ORS). The derived results show the proposed 2-RRS scheme’s performance compared to the conventional relay selection schemes in terms of system throughput, outage probability, and energy harvesting.The closed-form expressions of outage probability and average system throughput for the previous relay selections schemes and the proposed 2-RRS scheme are derived and validated by numerical simulation.The impact of multiantenna PB and other system parameters such as harvesting time, relays number, and position on the system performance are also investigated.

### 1.3. Organization of The Paper

The rest of this paper is organized as follows. Section 2 presents the system model and harvest-then-cooperate protocol. Section 3 introduces the conventional relay selection schemes, i.e., PRS, ORS, and our proposed 2-RRS relay selection scheme. In Section 4, we demonstrate the derivation of the outage performance and the system throughput of the considered system model with PRS, ORS, and 2-RRS relay selection schemes. Section 5 shows the numerical results and comparisons. Finally, the findings of the study are concluded in Section 6.

## 2. System Model

### 2.1. System Description

A multiantennas PB assisting a WPCCN is considered in this work, as shown in Figure 1. It consists of a source (*S*), a destination (*D*), and N>1 intermediate relay candidates ($R_{i}$), while $i\in \left\{1,2,3,...N\right\}$. All communications between *S* and *D* have to occur through the multirelay set due to the long-distance and the high path loss between them. *D* is powered by a fixed power supply, such as batteries. The PB is considered as an energy unconstrained node and transmits with a constant transmit power $P_{\mathrm{PB}}$ to *S* and the set of *N* relays node. Both *S* and the set of *N* relays node need to harvest the energy from the multiantenna PB beamformed signal in order to enable their information transfer, as it is assumed that they do not have an integrated power supply. Besides, *S*, $R_{i}$ and *D* nodes are equipped with one antenna and operate in half-duplex mode.

In the sequel, let ${\tilde{h}}_{AZ}∼\mathrm{CN}\left(0,\sigma_{AZ}^{2}\right)$ be the channel coefficient from *A* to *Z* where $A,Z\in \left\{S,R_{i},D\right\}$. The channel power gain from *A* to *Z* follows the exponential distribution, i.e., $h_{AZ}={\left|{\tilde{h}}_{AZ}\right|}^{2}∼\exp\left(\sigma_{AZ}^{2}\right)$ where the channel gain remains constant during each unit time interval (T) and may change from one time unit to another. We denote subscript-*B* for PB, assuming that PB has perfect knowledge of the channel vectors, let $G\in C^{(N+1)\times M}$ be the channel matrix whose *k*-th rows are the channel vectors form the PB array to *S* and *N* relay candidates $g_{BS}^{H}$, $g_{BR_{i}}^{H}$ respectively, i.e.,
(1)$$G^{H}≜{\left[g_{BS},g_{BR_{1}},g_{BR_{2}},...,g_{BR_{N}}\right]}^{H}.$$

The proposed system model uses TS-HTC protocol where the time block *T* is divided into three phases, and only one node transmits at one phase as depicted in Figure 2. We assume perfect CSI and synchronization in the network, but how to achieve this synchronization is beyond the scope of this paper. In the first phase, the PB beamform RF signal to *S* and $R_{i}$ nodes to allow them charge their batteries. In the second phase, by using the energy harvested in the 1st phase *S* transmit information to $R_{i}$. In the third phase, the best relay $R_{s}$ among the $R_{i}$ available relays is chosen to forward the information to *D* by either the new proposed 2-RRS relay selection scheme or the well-known conventional relay selection schemes, i.e., PRS and ORS, as described in Section 3.

Throughout this paper and in line with related works, the following set of assumptions are considered.

*A*_1_We follow location-based clustering model where the relays set are grouped relatively near each other. In the area of relay selection schemes this proposition is widely used (e.g., [10,11,25]) which means equal average channel power gains of B-R, S-R, and R-D links.*A*_2_The power beacon is considered as a dedicated power source for the network as proposed in [22,23]. PB, *S*, *R_i_*, *D* nodes are considered completely coordinated and run according to harvest and cooperate protocol. Uplink channel estimations without pilot contamination are assumed to be ideal, which employed for downlink ZF precoding computation based on channel reciprocity.*A*_3_As highlighted in [11,31], we assume that both the source and the relay candidates expend their amount of energy harvested in the transmission phases. Note that it is possible to consider power allocation methods here to enhance the efficiency of the system further. The above assumption is regarded as a lower bound of our proposed system model.

### 2.2. Signal Modeling

#### 2.2.1. Energy Beamforming Phase

First of all, in the first time slot τT with 0⩽τ<1, the PB equipped with *M* antennas beamform RF energy signal to both *S* and *N* relay candidates where N+1<M, let consider zero-forcing (ZF) processing given as
(2)$$v_{k}≜\sqrt{\beta_{k}M}{\left[G{\left(G^{H}G\right)}^{-1}\right]}_{:,k},$$
where ${\left[.\right]}_{:,k}$ is the *k*-th column of a matrix, $\beta_{k}$ is the path loss.

Thus, the received signal at *S* and *N* relays are respectively written as
(3)$$y_{S}=\sqrt{P_{\mathrm{PB}}}g_{BS}^{H}v_{k}s_{e}+w_{s},$$
(4)$$y_{R_{i}}=\sqrt{P_{\mathrm{PB}}}g_{BR_{i}}^{H}\sum_{k=2}^{K}v_{k}s_{e}+w_{r},$$
where $P_{\mathrm{PB}}$ denote the predefine transmit power of the PB during the harvesting time slot., $v_{k}$ is the beamforming vector related to the source and the *k*-th relay terminals where K=N+1, $s_{e}$ is the energy symbol with unit power, $w_{s}$ and $w_{r}$ are the complex Gaussian noise assumed to be a random variables with zero mean and unit variance, $w_{s},w_{r}∼\mathrm{CN}\left(0,1\right)$.

The PB equipped with multiantennas performs zero-forcing (ZF) precoder. This allows PB to design their beams to eliminate multiuser interference [6,8,32]. The multiuser interference term $\sum_{i=1,i\neq k}^{K}{\left|g_{k}^{H}v_{i}\right|}^{2}$ is zero, thus the harvested energy at the *S* and *N* relay candidates given by
(5)$$E_{S}=\eta \tau P_{\mathrm{PB}}{\left|g_{BS}^{H}v_{k}\right|}_{k=1}^{2},$$
(6)$$E_{R_{i}}=\eta \tau P_{\mathrm{PB}}\sum_{k=2}^{K}{\left|g_{BR_{i}}^{H}v_{k}\right|}^{2},$$
where 0<η<1 denotes the energy harvesting conversion efficiency factor, and τ is the duration time of the PB broadcasting the RF signal. ${\left|g_{BS}^{H}v_{k}\right|}^{2}$, ${\left|g_{BR_{i}}^{H}v_{k}\right|}^{2}$ are the channels gain between the PB and *S*, $R_{i}$ respectively.

#### 2.2.2. Information Transmission Phase

As depicted in Figure 2, in the remaining (1−τ)T time duration, *S* transmits the information to $R_{i}$ set using the energy collected in the previous harvesting phase. Finally, after one of the relay selection processes takes place, the best relay is selected to forward the *S* information to *D*. The transmit power at *S* and the $R_{i}$ relays set are thus written as
(7)$$P_{S}=\frac{E_{S}}{2(1-\tau )/3}=\frac{3\eta \tau P_{\mathrm{PB}}{\left|g_{BS}^{H}v_{1}\right|}^{2}}{2(1-\tau )}.$$
(8)$$P_{R_{i}}=\frac{E_{R_{i}}}{\left(1-\tau \right)/3}=\frac{3\eta \tau P_{\mathrm{PB}}\sum_{k=2}^{K}{\left|g_{BR_{i}}^{H}v_{k}\right|}^{2}}{1-\tau }.$$

Thus, the received Signal to Noise Ratio (SNR) from the source to the *N* relay candidates is expressed as follows
(9)$$\gamma_{SRi}=\frac{P_{S}h_{SRi}^{2}}{\sigma_{0}}$$

The received SNR from the *N* relay to the destination is written as
(10)$$\gamma_{RiD}=\frac{P_{R_{i}}h_{RiD}^{2}}{\sigma_{0}}$$
where 0<=i<=N, and $\sigma_{0}$ is the power of the noise associated with all receivers.

## 3. Relay Selection Schemes

In order to enhance system performance, we propose a novel relay selection scheme termed as 2-RRS. In this section, we compare the proposed 2-RRS with two popular relay selection schemes, i.e., PRS and ORS [11,25,26], as shown in Figure 3.

### 3.1. Opportunistic Relay Selection (ORS) Scheme

In ORS relay selection scheme both channel hops are important and should be taken into consideration [11,25,26]. Specifically, the best relay that maximizes the minimum of channels strengths between $\langle S\to R_{i}\rangle$ and $\langle R_{i}\to D\rangle$ is chosen and is given by
(11)$$R_{s}^{\mathrm{ORS}}=\arg\max_{i\in N}\left\{\min\left(\gamma_{SR_{i}},\gamma_{R_{i}D}\right)\right\}$$

### 3.2. Partial Relay Selection (PRS) Scheme

It is assumed in this scheme that the CSI is available for one hop only [11,25,26]. Particularly, when CSI is available for the first hop $\langle S\to R_{i}\rangle$ we denote the partial relay selection scheme in this case by (PRSI). In the case where the CSI is available only for the second hope $\langle R_{i}\to D\rangle$ we denote it by (PRSII), respectively. The selected relay in PRSI and PRSII can be expressed as
(12)$$R_{s}^{\mathrm{PRSI}}=\arg\max_{i\in N}\left\{\left(\gamma_{SR_{i}}\right)\right\}$$
(13)$$R_{s}^{\mathrm{PRSII}}=\arg\max_{i\in N}\left\{\left(\gamma_{R_{i}D}\right)\right\}$$

### 3.3. The Proposed Two-Round Relay Selection (2-RRS) Scheme

The proposed relay selection scheme is carried on at the destination, while both of energy harvesting and quality of the two-hop links are considered. Please note that the proposed relay selection includes two steps, namely, two rounds of selections: (1) choose the relay candidates, who successfully decode the source signal; (2) select the best relay among the relays chosen in the first round with the best R-D link condition to forward the source information to the destination as follows:By using the energy harvested in the first phase *S* deliver information to $R_{i}$ in the second phase. After $R_{i}$ receive *S* information each relay candidate sends his state information (the received SNR over $S\to R_{i}$ link and the amount of the EH from multiantennas PB) to *D*. The received SNR in each relay is compared with a predefined SNR threshold value $\gamma_{th}$. The *M* relays where 0⩽M⩽N which their SNR satisfy $\gamma_{SR_{i}}⩾\gamma_{th}$ are correctly decode *S* information and they will qualify for the second-round selection. We denote the selected group of relays as $\Omega_{j}^{M}$ and the best-selected relay as $R_{s}$.In the second selection round, the destination will choose the best of $\Omega_{j}^{M}$ relay group candidate as
(14)$$R_{s}^{2-\mathrm{RRS}}=\arg\max_{j\in M}\left\{\left(\gamma_{R_{j}D}\right)\right\}$$
where $\gamma_{R_{j}D}$ is the SNR of the jth relay candidate in the $\Omega_{j}^{M}$ to *D* via the link $\langle R_{j}\to D\rangle$, which can be written as
(15)$$\gamma_{RjD}=\frac{P_{R_{j}}h_{RjD}^{2}}{\sigma_{0}}$$The destination broadcasts the final choosing result to all relay candidates. Then, the selected optimal relay forwards the recoded information to the destination. The whole 2-RRS relay selection scheme is summarized with the following Algorithm 1:
**Algorithm 1:** Two-Round Relay Selection (2-RRS) Scheme Algorithm.**Input:** Define *N* number of relay, $\gamma_{th}$; **while**
*S* transmit to *N* Relays    Update $\gamma_{SR_{i}}$;**First Round Selection**  **for**
i=1:N
   Initialize $\Omega^{M}=0$;   **while**
$\gamma_{SR_{i}}>=\gamma_{th}$
   update the group $\Omega^{M}$;   There are *M* relay successfully decode *S* information, 0<M<N;   **end while**  **end for**
**Second Round Selection:**  **if**
M>0
**then**   Find the best relay $R_{s}$ maximizing $\gamma_{R_{j}D}$;   $R_{s}=\arg\max_{j\in M}\left(\gamma_{R_{j}D}\right)$  **end if**  **end while**



## 4. Performance Analysis

In this section, the system performance of the proposed 2-RRS and the conventional ORS and PRS schemes are analyzed. The closed-form outage probability and average system throughput of PB assisted WPCC networks are derived for all relay selection schemes mentioned above.

### 4.1. Outage Performance of the Proposed 2-RRS Scheme

In the proposed 2-RRS relay selection scheme both of energy harvesting and link quality of the two-hop links $\langle S\to R_{i}\rangle$ and $\langle R_{i}\to D\rangle$ are important. Let $\Omega_{j}^{M}$ be the group of *M* from *N* relays which successfully decodes the source information and satisfies the following condition
(16)$$\Omega_{j}^{M}=\left\{j:\gamma_{SRi}⩾2^{2R}-1\right\}$$
where *R* is the fixed transmission rate at the source. In the following let $\gamma_{th}=2^{2R}-1$.

After the group $\Omega_{j}^{M}$ is selected, the destination will choose the best relay candidate by selecting the $\max\left\{\gamma_{RjD}\right\}$. Thus, the outage probability of the proposed 2-RRS is mathematically written as
(17)$$P_{out}^{2\mathrm{RRS}}=\sum_{i=1,M=0}^{N}\sum_{j=1}^{\left({}_{M}^{N}\right)}\underset{\mathrm{Pr}\left(\Omega_{j}^{M}\right)}{\underset{︸}{\prod_{j\in \Omega_{j}^{M}}\mathrm{Pr}\left\{\gamma_{SRj}⩾\gamma_{th}\right\}\prod_{i\notin \Omega_{j}^{M}}\mathrm{Pr}\left\{\gamma_{SRi}⩽\gamma_{th}\right\}}}\underset{\mathrm{Pr}\left(\Omega_{j}^{M}|outage\right)}{\underset{︸}{\prod_{j\in \Omega_{j}^{M}}\mathrm{Pr}\left(\arg\max\left\{\gamma_{RjD}<\gamma_{th}\right\}\right)}}$$where $\mathrm{Pr}\left(\Omega_{j}^{M}\right)$ is the probability that the decoding group is selected, and $\mathrm{Pr}\left(\Omega_{j}^{M}|outage\right)$ is the probability that the selected relay group is in outage. Using the commutative propriety of multiplication
(18)$$P_{out}^{2\mathrm{RRS}}=\sum_{i=1,M=0}^{N}\sum_{j=1}^{\left({}_{M}^{N}\right)}\prod_{j\in \Omega_{j}^{M}}\left[\mathrm{Pr}\left\{\gamma_{SRj}⩾\gamma_{th}\right\}\mathrm{Pr}\left\{\gamma_{RjD}<\gamma_{th}\right\}\right]\prod_{i\notin \Omega_{j}^{M}}\mathrm{Pr}\left\{\gamma_{SRi}⩽\gamma_{th}\right\}$$

The wireless channels considered in this network are independent Rayleigh fading, and the channel gains follow the exponential random variables distribution, i.e., $h_{az}∼\exp\left(\sigma_{az}^{2}\right)$ [25]. Thus, the outage probability of the proposed scheme written as follows
(19)$$P_{out}^{2\mathrm{RRS}}=\sum_{i=1,M=0}^{N}\sum_{j=1}^{\left({}_{M}^{N}\right)}\left[\prod_{j\in \Omega_{j}^{M}}\left\{\underset{a_{j}}{\underset{︸}{\exp\left(-\frac{2^{R}-1}{2\eta P_{PB}h_{SRj}}\right)}}\left(1-\underset{b_{i}}{\underset{︸}{\exp\left(-\frac{2^{2R}-1}{2\eta P_{PB}h_{RjD}}\right)}}\right)\right\}\prod_{i\notin \Omega_{j}^{M}}\left(1-\underset{a_{i}}{\underset{︸}{\exp\left(-\frac{2^{R}-1}{2\eta P_{PB}h_{SRi}}\right)}}\right)\right]$$

By using the multinomial equality
(20)$$\begin{array}{cc} \sum_{M=0}^{N}\sum_{j=1}^{\left({}_{M}^{N}\right)}\prod_{i\in \Omega_{j}^{M}}\left[a_{i}\left(1-b_{i}\right)\right]\prod_{i\notin \Omega_{j}^{M}}\left[1-a_{j}\right] & =\prod_{k=1}^{K}\left[a_{k}\left(1-b_{k}\right)+\left(1-a_{k}\right)\right]\\ \; & =\prod_{k=1}^{K}\left[1-a_{k}b_{k}\right]. \end{array}$$

Thus, the outage probability closed form expression of the proposed 2-RRS relay selection scheme is expressed as follows
(21)$$P_{out}^{2\mathrm{RRS}}=\prod_{j=1}^{M}\left[1-\exp\left(-\frac{2^{R}-1}{2\eta P_{PB}h_{SRi}}\right)\exp\left(-\frac{2^{2R}-1}{2\eta P_{PB}h_{RiD}}\right)\right].$$

### 4.2. Outage Performance of the ORS Scheme

Interruption in $\langle S\to R_{i}\rangle$ or $\langle R_{i}\to D\rangle$ in two-hop DF transmission causes a complete system outage. Hence, in ORS, the system outage occurs when the best relay selected Rs fails to decode the source information or the destination cannot receive the forwarded information from Rs. Thus, the system outage probability in ORS scheme is written as
(22)$$\begin{array}{cc} P_{out}^{\mathrm{ORS}} & =\mathrm{Pr}\left(\max_{i\in N}\left\{\min\left(\gamma_{SRi},\gamma_{RiD}\right)<\gamma_{th}\right\}\right)\\ \; & =\mathrm{Pr}\left(\min\left(\gamma_{SR_{1}},\gamma_{R_{1}D}\right)<\gamma_{th}\right)\times \mathrm{Pr}\left(\min\left(\gamma_{SR_{2}},\gamma_{R_{2}D}\right)<\gamma_{th}\right)\times \cdots \times \mathrm{Pr}\left(\min\left(\gamma_{SR_{N}},\gamma_{R_{N}D}\right)<\gamma_{th}\right)\\ & =\prod_{i=1}^{N}\left[1-\mathrm{Pr}\left(\min\left(\gamma_{SR_{i}},\gamma_{R_{i}D}\right)>\gamma_{th}\right)\right] \end{array}$$

Follows [25] and from the fact that the minimum of two independent exponential random variables (r.v.’s) is again an exponential r.v. with a hazard rate equal to the sum of the two hazard rates. Thus, we obtain the outage probability for ORS as follows
(23)$$P_{out}^{\mathrm{ORS}}=\prod_{i=1}^{N}\left[1-\exp\left\{\frac{-v}{u}\left(\frac{1}{\sigma_{SRi}}+\frac{1}{\sigma_{RiD}}\right)\right\}\right].$$

### 4.3. Outage Performance of the PRS Scheme

Similar to [11], we can obtain the PRS outage probability as follows
(24)$$P_{out}^{\mathrm{PRS}}=\mathrm{Pr}\left(\gamma_{SRs}<\gamma_{th}\right)+\mathrm{Pr}\left(\gamma_{SRs}⩾\gamma_{th},\gamma_{RsD}<\gamma_{th}\right),$$

In the case of PRSI, the relay is selected based on the $\langle S\to R_{i}\rangle$ hop only. Follows the selection criteria $\gamma_{SRs}≜\max_{i\in N}\left\{\gamma_{SRi}\right\}$, the PRSI outage probability given as
(25)$$P_{out}^{\mathrm{PRSI}}=\prod_{i=1}^{N}\left[1-\exp\left(\frac{-v}{u\sigma_{SRi}}\right)\right]+\left[1-\exp\left(\frac{-v}{u\sigma_{RsD}}\right)\right]\left(1-\prod_{i=1}^{N}\left[1-\exp\left(\frac{-v}{u\sigma_{SRi}}\right)\right]\right)$$

When it comes to the PRSII protocol, only the $\langle R_{i}\to D\rangle$ link is available. Thus, the selection criteria should satisfy the expression, and the outage probability expression in this case is as follows
(26)$$P_{out}^{\mathrm{PRSII}}=\prod_{i=1}^{N}\left[1-\exp\left(\frac{-v}{u\sigma_{RiD}}\right)\right]+\left[1-\exp\left(\frac{-v}{u\sigma_{SRs}}\right)\right]\left(1-\prod_{i=1}^{N}\left[1-\exp\left(\frac{-v}{u\sigma_{RiD}}\right)\right]\right)$$

### 4.4. System Throughput

The effective transmission rate is $R\left(1-P_{out}^{*}\right)$, where *R* is the source fixed transmission rate and $P_{out}^{*}$ is the outage probability of derived earlier in this section. In this paper, we consider the delay-limited transmission mode [19], where the average throughput can be obtained by evaluating the outage probability of the system with a fixed transmission rate. Specifically, the system throughput is the product of the effective transmission rate and the information transmission time (1−τ) as depicted in Figure 1. Thus, the average system throughput is given as
(27)$$Th^{*}=R\left(1-P_{out}^{*}\right)\left(1-\tau \right)$$

The average system throughput is obtained by substitute $P_{out}^{*}$ with outage probability expression of each scheme. i.e., 2-RRS expression (21), ORS expression (23), PRSI expression (25), PRSII expression (26), are given respectively as follows
(28)$$\begin{array}{cc} \; & Th^{2-\mathrm{RRS}}=R\left(1-\prod_{j=1}^{M}\left[1-\exp\left(-\frac{2^{R}-1}{2\eta P_{PB}h_{SRi}}\right)\exp\left(-\frac{2^{2R}-1}{2\eta P_{PB}h_{RiD}}\right)\right]\right)\left(1-\tau \right) \end{array}$$
(29)$$\begin{array}{cc} \; & Th^{\mathrm{ORS}}=R\left(1-\prod_{i=1}^{N}\left[1-\exp\left\{\frac{-v}{u}\left(\frac{1}{\sigma_{SRi}}+\frac{1}{\sigma_{RiD}}\right)\right\}\right]\right)\left(1-\tau \right) \end{array}$$
(30)$$\begin{array}{cc} \; & Th^{\mathrm{PRSI}}=R\left(1-\prod_{i=1}^{N}\left[1-\exp\left(\frac{-v}{u\sigma_{SRi}}\right)\right]-\left[1-\exp\left(\frac{-v}{u\sigma_{RsD}}\right)\right]\left(1-\prod_{i=1}^{N}\left[1-\exp\left(\frac{-v}{u\sigma_{SRi}}\right)\right]\right)\right)\left(1-\tau \right) \end{array}$$
(31)$$\begin{array}{cc} \; & Th^{\mathrm{PRSII}}=R\left(1-\prod_{i=1}^{N}\left[1-\exp\left(\frac{-v}{u\sigma_{RiD}}\right)\right]-\left[1-\exp\left(\frac{-v}{u\sigma_{SRs}}\right)\right]\left(1-\prod_{i=1}^{N}\left[1-\exp\left(\frac{-v}{u\sigma_{RiD}}\right)\right]\right)\right)\left(1-\tau \right) \end{array}$$

## 5. Numerical Results and Discussion

In this section, MATLAB is used to run the Monte-Carlo simulation that validates our analytical results and derivations. In addition, the performance comparison between the proposed 2-RRS and the conventional PRS and ORS schemes are examined in terms of system throughput and outage probability. The impact of the multiantenna PB and other system parameters such as harvesting time, relay number, and position on the system performance are also provided.

Depending on a practical model of distance and path loss such as used in (e.g., [10,11,25]), we consider that $R_{i}$ are location-based clustering model and placed between *S* and *D* in one straightness, so the distance between *S* and $R_{i}$ is given as $d_{SRi}=d_{SD}-d_{RD}$. Without loss of generality, the system model parameters are defined in Table 1 unless otherwise stated.

Figure 4 and Figure 5 present the system throughput and the outage probability performance of the proposed 2-RRS scheme as a function of Power Beacon transmitted power $P_{\mathrm{PB}}\left(\mathrm{dB}\right)$, respectively. In order to examine the effect of energy beamforming on the system performance, a different PB antennas number is considered M={8,16,32,64}.

As we can depict from Figure 4, the throughput of the proposed 2-RRS scheme is enhanced when the number of PB antennas increases. Besides, the Monte-Carlo simulations match very well with our analytical results which validate our proposed model. There is an increment of about 0.28 pbs/Hz throughput with *M* growing from 8 to 16 when $P_{\mathrm{PB}}=15$ dB. However, when the PB power is above 15 dB and antennas number *M* increases above 32, the system tends to saturation. For instance, there is only a small throughput enhancement of 0.01 bps/Hz with *M* goes from 32 to 64 antennas when $P_{\mathrm{PB}}=15$ dB. These remarks match prior studies investigating the effect of the transmitter antennas number on the system performance [6,8,9], where the higher number of antennas at the transmitter side will be grantee more degree of freedom (DoF), which enhance the beamforming efficiency.

In addition, we can see from Figure 5 that there is an inverse proportion between the PB characteristics (transmit power $P_{\mathrm{PB}}$, antennas number *M*) and the outage probability $P_{out}$. Whereas, with $P_{\mathrm{PB}}$ and *M* increases in value, $P_{out}$ decreases significantly. For example, $P_{out}$ is around 0.1 with M=16 and $P_{\mathrm{PB}}=15$ dB, then $P_{out}$ dropped to only 0.001 when M=32 and $P_{\mathrm{PB}}=20$ dB.

In Figure 6 and Figure 7 we present the system throughput and outage probability performance comparison of the proposed 2-RRS scheme and that of the conventional ORS and PRS schemes versus PB transmitted power $P_{\mathrm{PB}}$. As we can see from Figure 6 and Figure 7 the proposed 2-RRS scheme outperforms the conventional relay selection schemes, i.e., ORS and PRS in terms of throughput and outage probability for both cases when M=8 and M=64. The simulation plots of all relay selection schemes are well-matched with the mathematical result, which validate the theoretical derivations of proposed 2-RRS and ORS, PRS selection schemes. Increases the number of PB antennas have a positive impact on all relay selection schemes. Table 2 provides quantitative performance comparison of the proposed 2-RRS over ORS and PRS [11,25,26] for the case when M=64. In the case of M=8 and $P_{\mathrm{PB}}$ takes small values (less than 13 dB), we can depict from Figure 6 that the throughput of all relay selection schemes tends to 0. This can be explained by the fact that all schemes are in an outage (outage probability around 1), as shown in Figure 7 and described in expression (27).

Table 2 provides numerical comparisons gain between our proposed 2-RRS scheme over the conventional ORS and PRS schemes [11,25,26]. The results are based on Figure 6 and Figure 7, when the number of PB antennas M=64 antennas. From Table 2 the proposed 2-RRS scheme has around 15% throughput gain over ORS scheme and approximately 37%, 62% throughput gain over PRSI and PRSII, respectively when $P_{\mathrm{PB}}=10$ dB. However, this throughput gain decreases to only 1% over ORS when $P_{\mathrm{PB}}=15$ dB and tends to 0 when $P_{\mathrm{PB}}=20$ dB. We can explain this degradation in gain by the saturation behavior when the SNR is high enough, which means the outage probability of all selection schemes tends to 0 in high PB transmit power. As in the case of $P_{\mathrm{PB}}=20$ dB the outage probability gain of the proposed scheme over ORS, PRSI, and PRSII are only (−0.0004, −0.029, −0.052), respectively.

Figure 8 demonstrates the impact of the number of relays on the system throughput performance with different relay selection schemes when $P_{\mathrm{PB}}=20$ dB and M=16. As can be observed from the curves, for each relay selection scheme, an optimized number of relays maximized the throughput performance. The proposed 2-RRS scheme has the best performance over the other schemes and achieves its optimum when N=6. Figure 8 revealed that it is not necessary the case that increasing the number of relays will lead to throughput performance enhancement, as the case with ORS, PRSI, and PRSII schemes, where the optimal number of relays for this schemes are N={5,4,3}, respectively. This remark can be justified by the surplus loss (M−N) in the DoF at the PB multiantennas when *N* is close to *M*.

In Figure 9, we investigate the impact of harvesting time allocation τ in the throughput system performance for different relay selection schemes when $P_{\mathrm{PB}}=10$ dB and M={16,64}. The optimal value of energy harvesting time for each relay selection scheme is obtained by an exhaustive search of the maximum system throughput. First of all, it can be seen from Figure 9 that all relay selection schemes required less harvesting time when PB is equipped with 64 antennas compared to M=16. This observation confirms previous result showing that increasing in PB antennas *M* will enhance the system performance. Second, to achieve the maximum throughput performance, our proposed 2-RRS scheme required relatively less harvesting time (0.1 s less) than all the compared schemes. Specifically, the optimum harvesting time for the proposed scheme is τ=0.3 s when M=64, and τ=0.6 s when M=16. Third, in Figure 9 we can see from the relay selection schemes curves that there is a tradeoff between the fraction of time τ allocated for energy harvesting, and the remaining time (1−τ) reserved for information transmission. Although a larger value of τ means more harvested energy at *S*, Ri and leads to less outage probability, the small portion of time remaining for information transmission will infect the throughput performance as it is shown in expression (27).

Channels in wireless power transfer are determined not only by the reception reliability but also by the attenuation of RF source transmission power, which is fundamentally different from conventional cooperative networks. This difference makes the distances between the PB and harvested nodes (*S* and *R_i_*) more critical. In order to study the impact of the PB localization between *S* and *R_i_* on the system performance. We change the position of the PB $\left(d_{\mathrm{PB}}\right)$ and based on Pythagoras’s theorem; we calculate how *S* and *R_i_* are far from the PB. (i.e., $d_{B}S$ and $d_{BRi}$), respectively, as shown in Figure 10.

The result in Figure 11 illustrates the position of the PB between *S* and *R_i_* and its influence on the system throughput for different relay selection schemes $P_{\mathrm{PB}}=15$ dB and M=64. The proposed 2-RRS scheme always possesses the best throughput performance over the other schemes regardless of the PB location $d_{\mathrm{PB}}$. Moreover, the optimal PB location that maximized the proposed 2-RRS throughput is when $d_{\mathrm{PB}}=2.5$ m similar to PRSI scheme. Compared to the proposed 2-RRS and PRSI schemes, the optimal PB position for ORS and PRSII schemes is relatively good when PB deployed nearer to the source. This observation is understandable since the CSI of the S-R links is not available for PRSII scheme. In the case of ORS scheme, the S-R links are more critical than the R-D links; this is because when the S-R link is in an outage, all the system will suffer from an outage due to max-min criteria applied in this scheme. Finally, from Figure 6, Figure 7, Figure 8 and Figure 9, Figure 11, it is worth arguing that the proposed 2-RRS relay selection scheme is given the best system performance in all considered cases superior to the compared schemes ORS, PRSI, and PRSII.

## 6. Conclusions

In this paper, a PB assisted wireless power cooperative communication network (PB-WPCCN) is considered. We proposed a two-round relay selection scheme (2-RRS). In the first round, a group of relays that successfully decoded the source information is selected. In the second round, the best relay among the selected group forwards the information to the destination. We compared the new proposed 2-RRS scheme with two popular relay selection schemes: opportunistic relay selection (ORS) and partial relay selection (PRS). The outage probability and the average system throughput closed-form expressions of the new proposed scheme and the conventional schemes are derived over a Rayleigh fading channel. The numerical results have shown the supercity of the proposed relay selection scheme which guarantee {15%,37%,62%} of throughput gain over the compared ORS, PRSI and PRSII schemes, respectively when the PB parameters are $P_{\mathrm{PB}}=10$ dB and M=64. When more PB antennas are added, the energy harvesting at source and relays set increases due to excess in DoF. The proposed scheme’s outage performance enhanced with nearly 69% over ORS scheme and {82%,87%} over PRSI PRSII schemes, respectively when $P_{\mathrm{PB}}=10$ dB. Our proposed 2-RRS scheme required (0.1 s) less harvesting time than other schemes. The optimal energy harvesting time values are obtained by an exhaustive search of the maximum system throughput for each relay selection scheme when M={16,64}. It is noteworthy that increasing relays number and harvesting time does not necessarily positively impact the throughput performance; thus, tradeoffs between energy harvesting and information transmission have been considered. Finally, choosing the best PB position depends on the selection criteria for each relay selection scheme, where $d_{\mathrm{PB}}=2.5$ m is the optimal PB distance for the proposed 2-RRS and PRSI schemes and $d_{\mathrm{PB}}=\left\{1.5,0.5\right\}$ m are the optimal PB location for ORS and PRSII schemes, respectively.

## Figures and Tables

**Figure 1 sensors-21-00147-f001:**
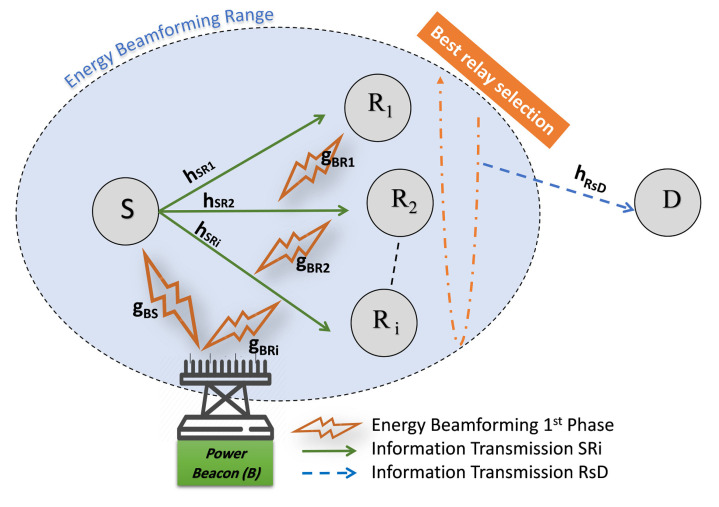
Multiantennas Power beacon-assisted wireless-powered cooperative communication network (PB-WPCCN) with energy beamforming and cooperative information transmission.

**Figure 2 sensors-21-00147-f002:**
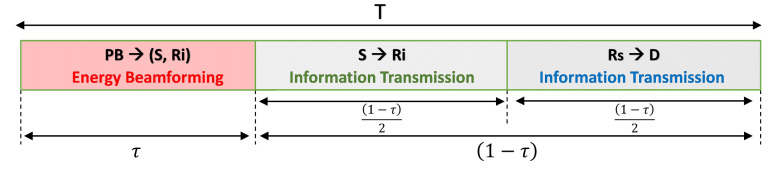
Diagram of the time switching harvest then cooperate protocol (TS-HTC).

**Figure 3 sensors-21-00147-f003:**
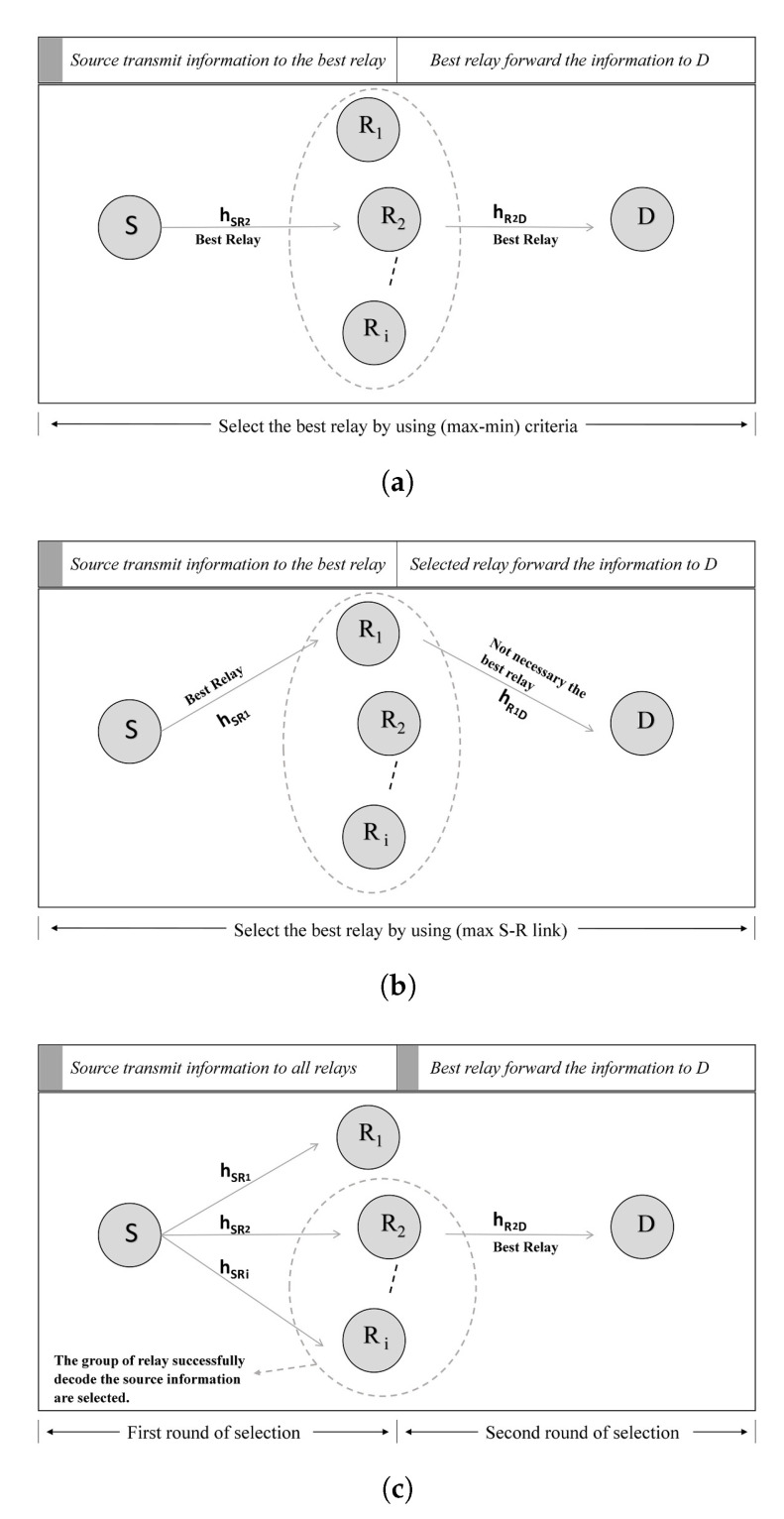
A dual-hop half-duplex decode and forward relay selection schemes. The grey shaded band shows when relay selection occurs. (**a**) Opportunistic Relay Selection(ORS) scheme [11,25,26]; (**b**) Partial Relay Selection (PRS) scheme [11,25,26]; (**c**) The proposed Two-Round Relay Selection (2-RRS) scheme.

**Figure 4 sensors-21-00147-f004:**
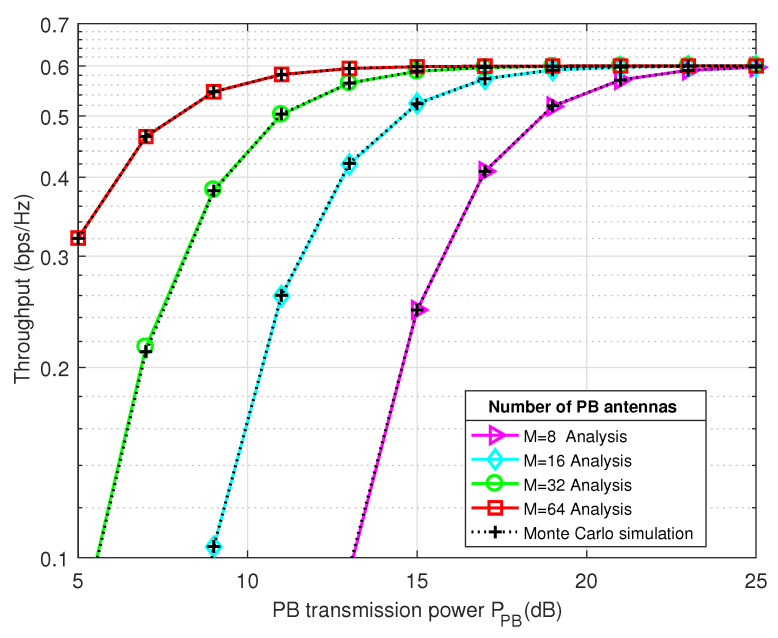
The system throughput performance of the proposed 2-RRS scheme versus the Power Beacon transmitted power $P_{\mathrm{PB}}\left(dB\right)$ for different PB antennas number M={8,16,32,64}.

**Figure 5 sensors-21-00147-f005:**
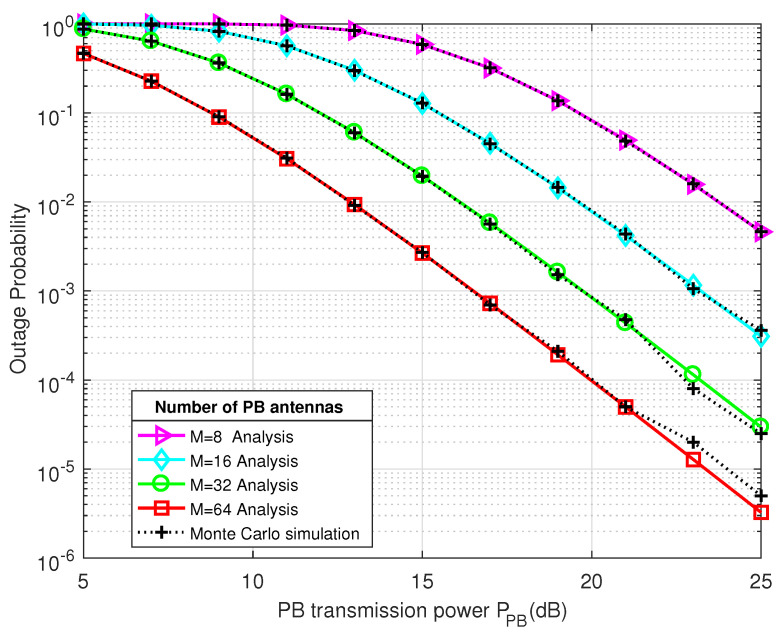
The outage probability of the proposed 2-RRS scheme versus the Power Beacon transmitted power $P_{\mathrm{PB}}$ (dB) for different PB antennas number M={8,16,32,64}.

**Figure 6 sensors-21-00147-f006:**
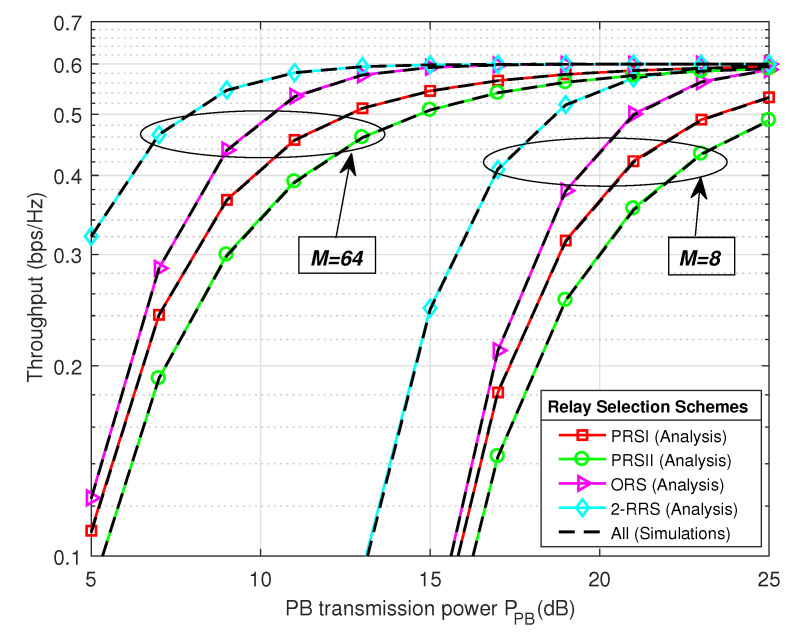
The system throughput performance comparison of the proposed 2-RRS scheme and that of the conventional ORS and PRS versus Power Beacon transmitted power $P_{\mathrm{PB}}$ (dB) with M={8,64}.

**Figure 7 sensors-21-00147-f007:**
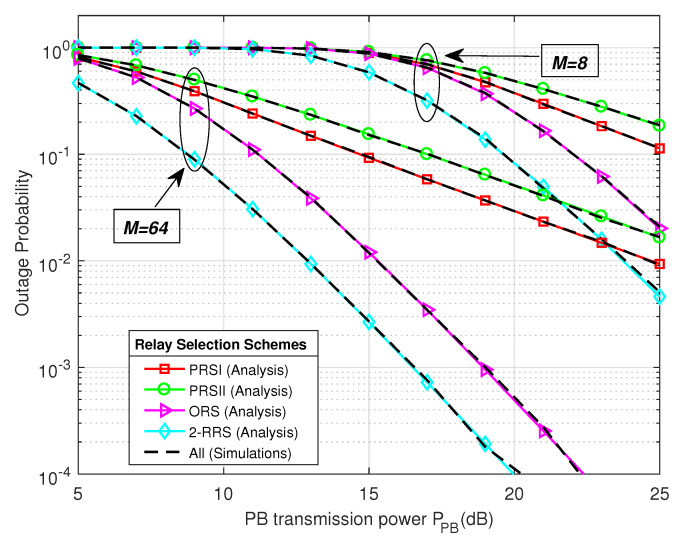
The outage probability comparison of the proposed 2-RRS scheme and the conventional PRS and ORS schemes versus the power beacon transmit power $P_{\mathrm{PB}}$ with M=8 and M=64.

**Figure 8 sensors-21-00147-f008:**
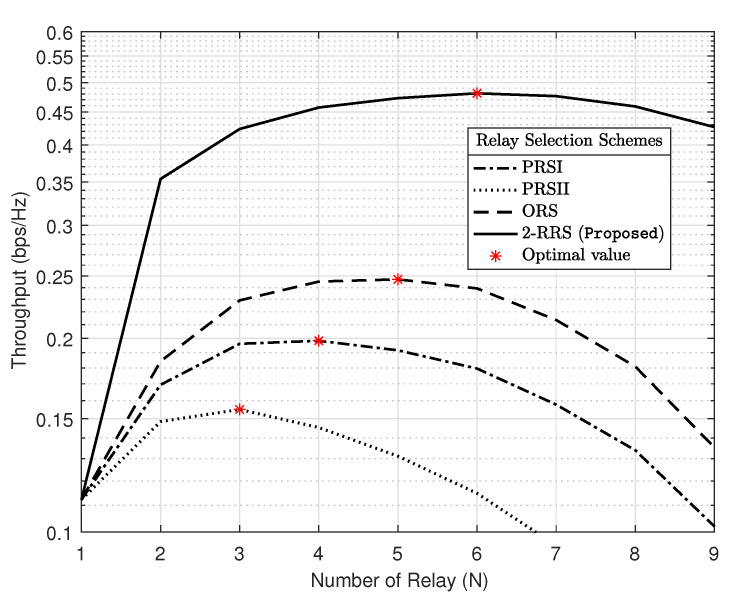
The impact of number of relays on the system throughput performance with different relay selection schemes with $P_{\mathrm{PB}}=20$ dB and M=16.

**Figure 9 sensors-21-00147-f009:**
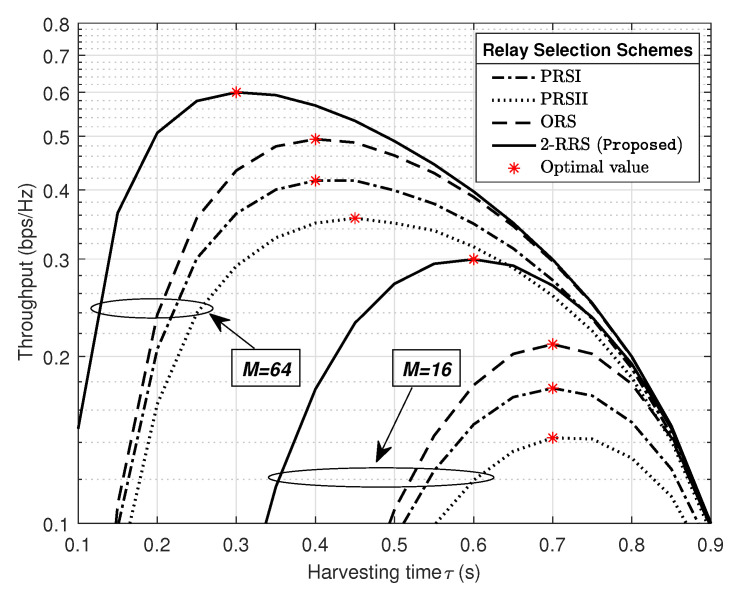
The impact of harvesting time allocation τ in the throughput system performance for different relay selection schemes with $P_{\mathrm{PB}}=10$ dB and M={16,64}.

**Figure 10 sensors-21-00147-f010:**
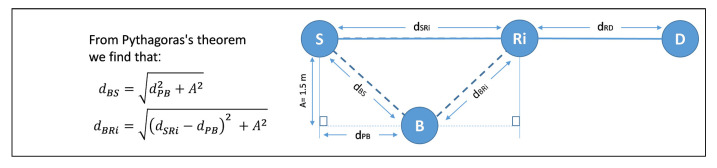
Localization of the PB between *S* and *R_i_*.

**Figure 11 sensors-21-00147-f011:**
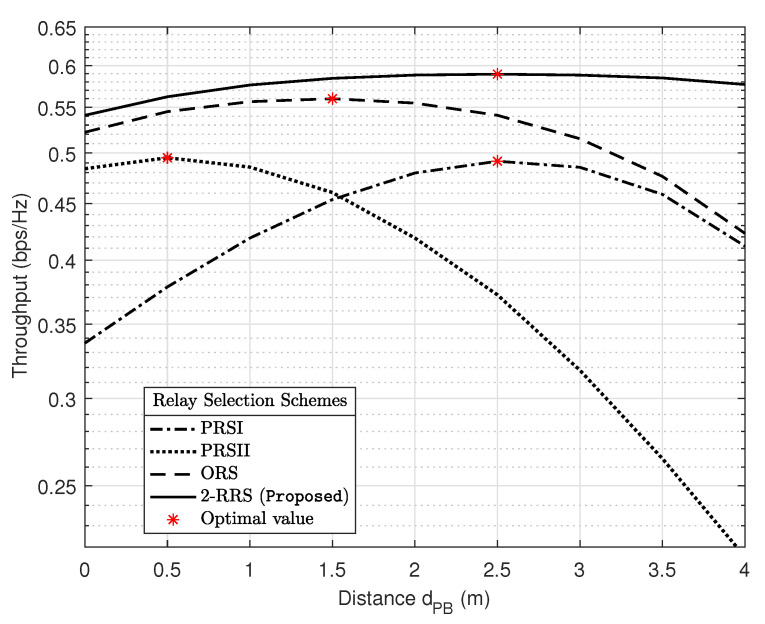
The influence of the PB localization on the system throughput for different relay selection schemes with $P_{\mathrm{PB}}=15$ dB and M=64.

**Table 1 sensors-21-00147-t001:** System parameters used in the performance evaluation.

System Parameter	Value
The number of relays	N=3
The number of antennas of PB	M=8,16,32,64
The transmission rate of the source	R=2 bps/Hz
Energy conversion efficiency	η=0.7
Time block	T=1 s
Harvesting time	τ=0.4 s
Distance from S-Ri	$d_{SRi}=4$ m
Distance from S-D	$d_{SD}=7$ m
Distance from PB-S and PB-Ri	$d_{BS}=d_{BRi}=2.5$ m
Noise power	$\sigma_{0}=1$ [33]
The fading severity	m=2

**Table 2 sensors-21-00147-t002:** Throughput and Outage probability of our proposed 2-RRS scheme over the ORS and PRS relay selection schemes [11,25,26] with M=64 antennas.

	Throughput (bps/Hz)			Outage Probability		
	**2-RRS (Proposed) Over ORS [11,25,26]**	**2-RRS (Proposed) Over PRSI [11,25,26]**	**2-RRS (Proposed) Over PRSII [11,25,26]**	**2-RRS (Proposed) Over ORS [11,25,26]**	**2-RRS (Proposed) Over PRSI [11,25,26]**	**2-RRS (Proposed) Over PRSII [11,25,26]**
$P_{\mathrm{PB}}=10$ dB	+0.0742 (15.02%)	+0.1533 (36.95%)	+0.2182 (62.34%)	−0.1224 (69.57%)	−0.2543 (82.61%)	−0.3653 (87.22%)
$P_{\mathrm{PB}}=15$ dB	+0.0055 (0.93%)	+0.0536 (9.84%)	+0.0909 (17.91%)	−0.0095 (78.20%)	−0.0896 (97.12%)	−0.1510 (98.27%)
$P_{\mathrm{PB}}=20$ dB	+0.0002 (0.03%)	+0.0176 (3.02%)	+0.0306 (5.38%)	−0.0004 (77.66%)	−0.0287 (99.64%)	−0.0520 (99.80%)

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
