# Peer review of "Optimal Relay Selection Scheme with Multiantenna Power Beacon for Wireless-Powered Cooperation Communication Networks"

_sensors, 2020, doi:10.3390/s21010147_

Round 1

Reviewer 1 Report

The topic is interesting; the algorithm proposed is novel, and the simulation shows a clear system improvement brought by the protocol proposed.

However, find below few issues that have to be addressed:

On line 32 there is a lack of w in the statement "ireless-powered communication (WPC)"

In table 1: the is a comma missed after 32 (the second line) and between the value and the measurement unit should be a space (for example 4m or 7m).

On line 242, the sentence: 

"As we can depict from Figure 4 that the throughput of the proposed 2-RRS scheme is enhanced when the number of PB antennas increases."

has more sense as:

As we can depict from Figure 4, the throughput of the proposed 2-RRS scheme is enhanced when the number of PB antennas increases.

 On line 247, described has more sense than describe.

On line 289, I believe you meant "farthest" not fares.

In conclusions, line 316, you should remove "is" from the statement:

Also, choosing the best PB position is depends on the selection criteria 316
for each relay selection scheme.

On line 317, I believe the authors meant:

On the other hand, when more PB antennas are added, the energy harvesting at source and relays set increases, which positively impacts the system outage performance.

Author Response

We thank the referee for the insightful comments on our manuscript.

Concern # 1: On line 32 there is a lack of w in the statement "ireless-powered communication (WPC)"

Author action: We updated the manuscript by adding the letter “w” to “wireless-powered communication (WPC)”. Line 43

Concern # 2: In table 1: the is a comma missed after 32 (the second line) and between the value and the measurement unit should be a space (for example 4m or 7m).

Author action: We updated Table 1 in the manuscript by adding comma btw (32, 64) and spaces where relevant.

Concern # 3: On line 242, the sentence: "As we can depict from Figure 4 that the throughput of the proposed 2-RRS scheme is enhanced when the number of PB antennas increases."

has more sense as:

As we can depict from Figure 4, the throughput of the proposed 2-RRS scheme is enhanced when the number of PB antennas increases.

Author action: As suggested by reviewer we change the sentence “As we can depict from Figure 4 that the throughput of the proposed 2-RRS scheme is enhanced when the number of PB antennas increases.” to “As we can depict from Figure 4, the throughput of the proposed 2-RRS scheme is enhanced when the number of PB antennas increases”. Line 318-319

Concern # 4:

On line 247, described has more sense than describe.

The manuscript is updated by the reviewer suggested word “described”. Line 342

On line 289, I believe you meant "farthest" not fares.

Thank you for pointing this out, it was a typo mistake we meant “far”, the manuscript is updated in Line 381

In conclusions, line 316, you should remove "is" from the statement:

Also, choosing the best PB position is depends on the selection criteria for each relay selection scheme.

Thank you for pointing this out, we removed “is” from the sentence. Line 412

Concern # 5: On line 317, I believe the authors meant:

On the other hand, when more PB antennas are added, the energy harvesting at source and relays set increases, which positively impacts the system outage performance.

Author action: We change the sentence “On the other hand, adding more PB antennas increases the energy harvesting at source and relays set, which positively impacts the system outage performance.” To the sentence “When more PB antennas are added, the energy harvesting at source and relays set increases”. Line 404-405

Reviewer 2 Report

The authors propose an investigation about multi-antennas power beacon to improve the wireless-powered cooperative communication network. The authors propose a new relay selection scheme, named two-round relay selection, where a group of relays successfully decode the source information is selected in the first round selection. In the second round, the optimal relay is selected to forwards the recorded information to the destination. The authors compare his/her technique with partial relay selection (PRS) and opportunistic relay selection (ORS), obtaining better results in terms of throughput and power transmission.

The topic is relevant and actual, especially the possibility to be applied to IoT environments.

Related work is present together with the Introduction. It is necessary to create a dedicated section for related work.

The authors need to improve and justify the state of the art. 

ORS and PRS techniques need to be better introduced. It is not clear the difference between them.

Some typos need to be solved: as "new ireless-powered communication (WPC)".

Author Response

We thank the reviewer for this positive feedback and his valuable comments.

Please find the corrections and responses in the attached file. 

Reviewer 3 Report

The paper presents a relay selection scheme with multi-antenna power beacon for wireless-powered cooperation communication networks. The authors use the Monte-Carlo simulation in MATLAB for their research.

The manuscript should be carefully reviewed, clarifications and additions are necessary.

The numerical results and their interpretation are not presented clearly enough.

Figures 4, 6, 8, 9, 11: The unit of measurement on the vertical axis is missing.

Figure 9: The unit of measurement on the horizontal axis is missing.

The conclusions should be rewritten and more focused on the results obtained and not just general discussion. The authors should present comparatively energy efficiency, network capacity and reception reliability for the proposed scheme (quantitative data not only general assessments).

Author Response

We thank the reviewer for his careful thoughts and constructive suggestions to improve our manuscript.

Please find the attached document contains responses and corrections. 

Round 2

Reviewer 1 Report

Thank you for considering my suggestions and improved your work. I wish you to have successful research work in this area. 

Wish you a Happy Holliday season and healthy days!

Reviewer 3 Report

My comments have been addressed.